# Generating fast-twitch myotubes in vitro with an optogenetic-based, quantitative contractility assay

Katharina Hennig[1], David Hardman[4], David MB Barata[1], Inês IBB Martins[1], Miguel O Bernabeu[4,5], Edgar R Gomes[1], William Roman[1,2,3]

**The composition of fiber types within skeletal muscle impacts the tissue's physiological characteristics and susceptibility to disease and ageing. In vitro systems should therefore account for fiber-type composition when modelling muscle conditions. To induce fiber specification in vitro, we designed a quantitative contractility assay based on optogenetics and particle image velocimetry. We submitted cultured myotubes to long-term intermittent light-stimulation patterns and characterized their structural and functional adaptations. After several days of in vitro exercise, myotubes contract faster and are more resistant to fatigue. The enhanced contractile functionality was accompanied by advanced maturation such as increased width and up-regulation of neuron receptor genes. We observed an up-regulation in the expression of fast myosin heavy-chain isoforms, which induced a shift towards a fast-twitch phenotype. This long-term in vitro exercise strategy can be used to study fiber specification and refine muscle disease modelling.**

## Introduction

Skeletal muscle tissue possesses distinct muscle fiber types, which are broadly categorized by their contractile properties: slow (type I) and fast (type II) twitch muscle cells (myofibers). The proportion of fiber types depends on the functional demand of specific muscles (Agbulut et al, 2003). Slow, fatigue-resistant fibers are predominant in muscles involved in posture, whereas fast-twitching fibers possess a higher mechanical force output and are present in muscles devoted to motion (Schiaffino & Reggiani, 2011). Moreover, muscle composition can adapt through fiber-type switching in response to internal and external factors, such as neuromuscular activity, exercise or hormone exposure (Blaauw et al, 2013). Ageing, injury, and disease have also been linked to changes in fiber-type composition (Jansen & Fladby, 1990). For example, sarcopenia, the

age-related loss of muscle mass, is characterized by a loss of type II-associated satellite cells (Frontera & Ochala, 2015) whereas nerve denervation, which occurs in certain neuromuscular disorders, leads to a slow-to-fast fiber switch (Ciciliot et al, 2013; Peggion et al, 2017). Muscle disorders also tend to predominantly affect one muscle type over another such as in myotonic dystrophy type 1 and type 2 in which type I and II myofibers are more susceptible to damage, respectively (Talbot & Maves, 2016).

Despite the importance of fiber type in disease pathophysiology, fiber type is rarely considered when modelling muscle disorders in vitro. Advances in tissue engineering and in vitro cell models have offered new tools to better mimic native muscle tissue and muscle disorders. Various techniques have contributed to enhance the structural maturity and contractile functionality of artificial muscles such as (i) cell alignment via microfabrication (Zhao et al, 2009; Bajaj et al, 2011), (ii) 3D culture systems (Maffioletti et al, 2018), (iii) exercise-like stimuli (Asano et al, 2015; Sebille et al, 2017; Aguilar-Agon et al, 2019), and (iv) integration of other cell types (e.g., innervating neurons [Osaki et al, 2018b; Bakooshli et al, 2019; Guo et al, 2020] and vasculature [Osaki et al, 2018a]). The validation of improved myogenesis relies on a combination of molecular, morphological, and functional assays. However, most studies do not perform any characterization of fiber-type composition. As these in vitro systems are increasingly adopted for drug screening and biological investigation, the systematic characterization of muscle tissue composition is crucial, especially to model diseases affecting distinct fiber types.

The expression of distinct myosin isoforms determines fiber type and contractile properties. Myosin consists of two myosin heavy chains (Myh), two regulatory light chains, and two essential chains. Myh with its ATPase activity and actin-binding domain is responsible for power strokes during contractions. Interestingly, ATPase activity of distinct Myh isoforms is directly linked with the myofiber's contraction velocity (Bottinelli et al, 1991, 1994). Hence, Myh isoforms are the most appropriate biomarkers to characterize fiber types. During development, the expression of Myh isoforms follows a sequential, temporal pattern (Agbulut et al, 2003): developmental

[1]Instituto de Medicina Molecular, Faculdade de Medicina, Universidade de Lisboa, Lisboa, Portugal [2]Australian Regenerative Medicine Institute, Monash University, Clayton, Australia, [3]Victoria Node, EMBL Australia, Clayton, Australia [4]Centre for Medical Informatics, Usher Institute, The University of Edinburgh, Edinburgh, UK [5]The Bayes Centre, The University of Edinburgh, Edinburgh, UK

Correspondence: william.roman@monash.edu; edgargomes@medicina.ulisboa.pt

Myh, namely embryonic Myh3 (emb-Myh3) and neonatal Myh8 (neo-Myh8), is expressed together with slow Myh7 (slow-Myh7). Throughout the course of differentiation, developmental Myh isoforms disappear when slow-Myh7 or fast-Myh isoform (fast-Myh1, fast-Myh2, and fast-Myh4) expression is up-regulated. The predominantly expressed Myh isoform determines the adult fiber type and its associated contraction dynamics. Compared with adult Myh isoforms, embryonic Myh isoforms possess slower activation and relaxation kinetics and a lower rate of force production (Racca et al, 2013), whereas adult fibers that predominantly express slow Myh7 contract with a slower velocity than fibers containing fast-Myh isoforms (Pellegrino et al, 2003; Johnson et al, 2019).

In this study, we describe an experimental strategy to alter fiber-type composition of in vitro muscle systems. By subjecting myotubes to long-term, intermittent mechanical training, we monitored structural, functional, and molecular changes underlying fiber-type switching. Long-term in vitro exercise alters the temporal protein expression pattern of Myh isoforms: trained myotubes down- and up-regulate the expression of slow Myh7 and fMyh, respectively. Our results suggest that rhythmic, long-term mechanical training triggers a phenotypical shift toward a fatigue-resistant, fast fiber type. This approach could be used to model fiber type-specific diseases and identify new therapeutic targets.

# Results

### Designing a quantitative, optogenetics-based in vitro contractility assay

We aimed to determine if long-term exercise of in vitro myotubes alters contraction dynamics. To do so, we relied on an in vitro muscle protocol, which generates highly differentiated myotubes (Falcone et al, 2014; Pimentel et al, 2017). After 7 d of differentiation, these myotubes exhibit hallmarks of mature muscle cells, such as peripheral nuclei, striations, and contractility. To control their contraction, we infected these cultures with the AAV9-pACAGW-ChR2-Venus (adenovirus) at day 0, resulting in a high percentage of myotubes (90.38%; Fig S1A and B) expressing ChR2 at day 4. We confirmed the cells' functional photosensitivity by exposing ChR2-expressing myotubes to blue light under an inverted fluorescent microscope. This resulted in 86.99% of total myotubes performing cycles of contraction under continuous illumination.

To submit myotubes to specific and long-term stimulation patterns, we designed an optogenetic-based, in vitro training platform (termed OptoPlate; Fig 1A and B). The OptoPlate emits blue light pulses with high temporal control for extended durations and is compatible with cell culture. The device consists of a 3D-printed six-well plate made of polylactic acid with integrated blue LEDs (475 nm). Seven LEDs are grouped under each well and can be controlled independently via an Arduino board. A touch-screen interface allows setting training parameters (e.g., pulse length, light intensity, and duration) for each 35 mm dish separately.

To measure contraction dynamics after long-term training, we designed a quantitative, functional contractility assay based on particle image velocimetry (PIV) analysis. We experimentally assessed contractile behaviors at day 4 of differentiation (after 3 d of training) by continuously illuminating ChR2-expressing myotubes under the microscope and recording contractions for 2 s (Fig 1C). We used PIVlab, an imaged-based PIV analysis software (Thielicke & Stamhuis, 2014), to analyze contraction kinetics quantitatively. The program divides images into interrogation windows, computes the local displacement of two consecutive windows via maximum correlation methods, and outputs vector fields and velocity magnitudes (Fig 1D). We use both these measurements to determine contraction kinetics such as myotube displacement and contraction velocities of single myotubes (Fig 1E and F). Finally, we monitored the time until myotubes enter fatigue by continuously stimulating cultures with blue light until they stop responding (Fig 1G). We employed this optogenetic contractility assay to evaluate the effect of long-term exercise on muscle cell function.

### Long-term in vitro exercise accelerates myotube contraction speed and decreases fatigability

The OptoPlate allows submitting myotubes to distinct long-term exercise programs by setting specific light-stimulation parameters (intensity, duration, and frequency). To trigger contractions in a noninvasive manner with high temporal control, myotubes were infected with AAV9-pACAGW-ChR2-Venus at day 0 of differentiation (Fig 2A). We first tested if an optimal stimulation frequency enhances contractile properties of myotubes. Three OptoTraining protocols were tested: 50-ms light pulses at 2, 5 or 10 Hz (Fig 2B). To mimic chronic exercise, we trained myotubes for 8 h per day. By using low-light intensity, we minimized any phototoxicity effect. In addition, long-term OptoTraining did not cause any sarcomeric damage. Although both untrained and trained myotubes exhibit local sarcomeric scars visualized via filamin C staining (Roman et al, 2021), they display no significant increase in filamin C mRNA or protein expression (Fig S2A–F).

Using our PIV-based contractility assay, we investigated functional adaptations of myotubes to the different OptoTraining protocols. Interestingly, myotube contraction velocity positively correlates with light stimulation frequency: higher stimulating frequencies resulting in faster myotube twitching after training (Fig 2C). However, 10 Hz stimulated myotubes are more prone to fatigue (Fig 2D) for a marginal increase in contraction velocity when compared with the 5 Hz cohort. As such, we selected 5 Hz as the optimal training frequency. All further experiments compare untrained control myotubes to the 5 Hz OptoTraining protocol (from here on generally referred to as "trained myotubes").

Individual trained myotubes possess faster contraction cycles compared with untrained control myotubes (Fig 2E–G). To understand the origin of this increased contraction speed, we performed a deeper quantitative analysis of contraction kinetics. For this, we developed an algorithm dissecting individual displacement curves of contracting myotubes into acceleration and relaxation phases (Fig 2H) and observed significantly shorter acceleration (Fig 2I) and relaxation phases for the trained myotubes (Fig 2J). These trained myotubes therefore undergo contractions with higher frequency and shorter wavelength leading to faster contraction cycles (Fig 2K and L).

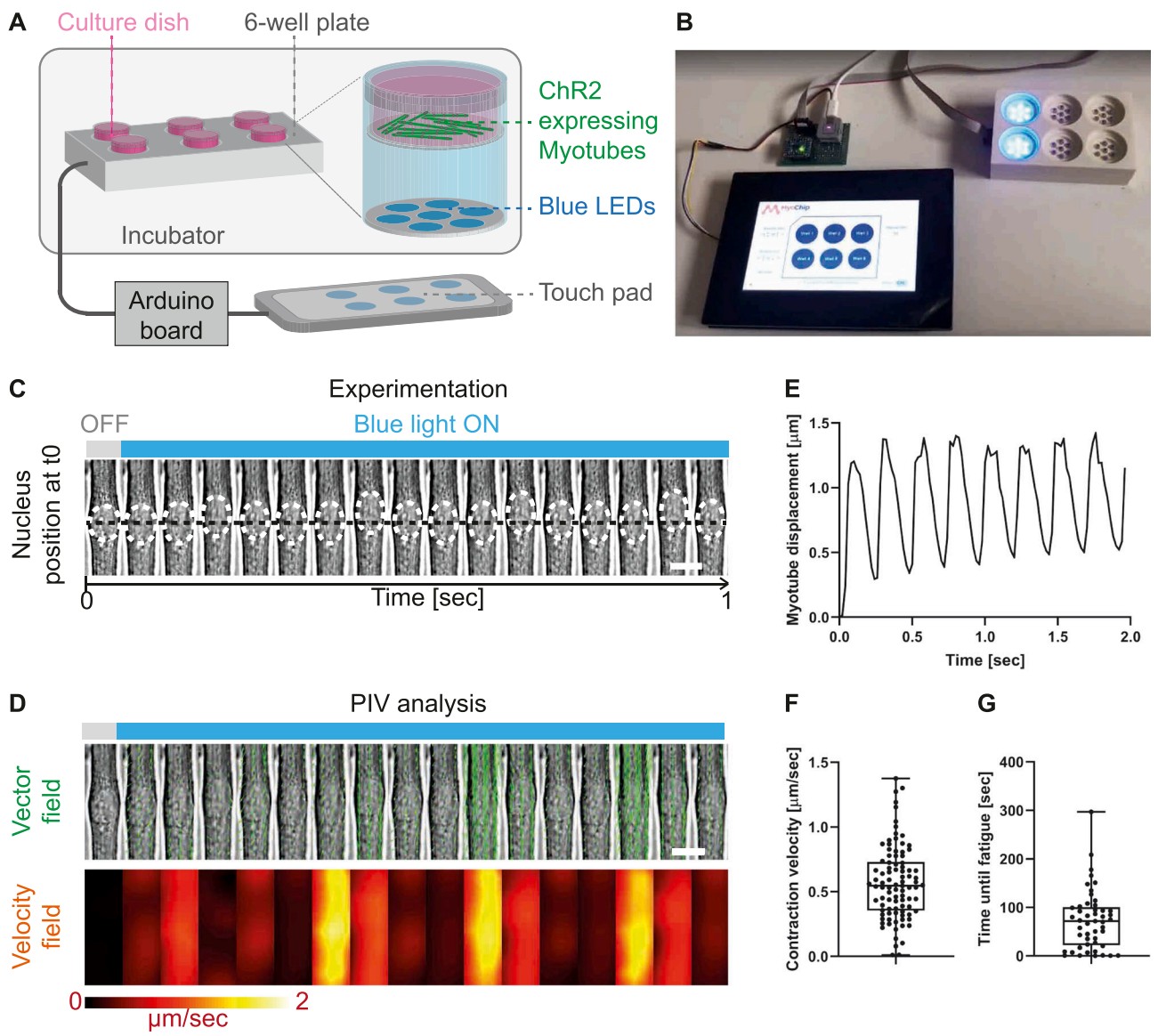

**Figure 1. Strategy to submit myotubes to long-term light-induced exercise and quantify contraction kinetics with single-cell resolution.**
**(A, B)** Schematic illustration (A) and image of designed OptoPlate (B). **(C)** Kymograph of a contracting 4-d myotube continuously stimulated with blue light. White dashed circles highlight the nucleus and the black dashed line marks the initial nuclear position before stimulation. **(D)** Particle image velocimetry analysis applied to myotube displacement over time. Displacement vectors (green arrows) and velocity fields (heat map) convey myotube movement. **(E)** Representative displacement curve of a single-contracting myotube. **(F)** Boxplot of myotube contraction velocity averaged over 2 s (n = 100). **(G)** Boxplot of time needed for cells to enter fatigue when continuously light stimulated (n = 49). Data information: (F, G) experiments performed in triplicates and outliers were identified using ROUT method (Q = 1). Scale bars: (C, D) 10 μm.

## OptoTraining increases cell width and promotes synaptogenesis

Because contraction depends on muscle architecture, we sought to understand if changes in cellular structures underlie the faster contraction cycles observed after long-term in vitro training. We performed a quantitative structural analysis by extracting parameters of myotube maturation from immunofluorescent images at day 4 of differentiation (Fig 3A and B; Hardman et al, 2021 *Preprint*). Trained myotubes display an increased cell width (Fig 3C), suggesting an addition of sarcomeres because of training. We also observed shorter myonuclear distances (Fig 3D), whereas the

myonuclei variability coefficient (a measure of how uniformly myonuclei are distributed within myotubes) remains unchanged (Fig 3E). These maturation trends are conserved for 2 and 10 Hz stimulation frequencies, albeit more discrete (Fig S3A–C). Surprisingly, we did not observe a significant difference in the percentage of striated myotubes at day 4 (Fig 3F), suggesting that initiation of myofibril assembly is similar in trained and untrained cultures. However, Western blot analysis shows an increased protein expression of α-actinin for trained myotubes which persists over 7 d (Fig 3G–I). Nevertheless, we did not find a correlation between fiber width and contraction speed (Fig 3J), indicating that

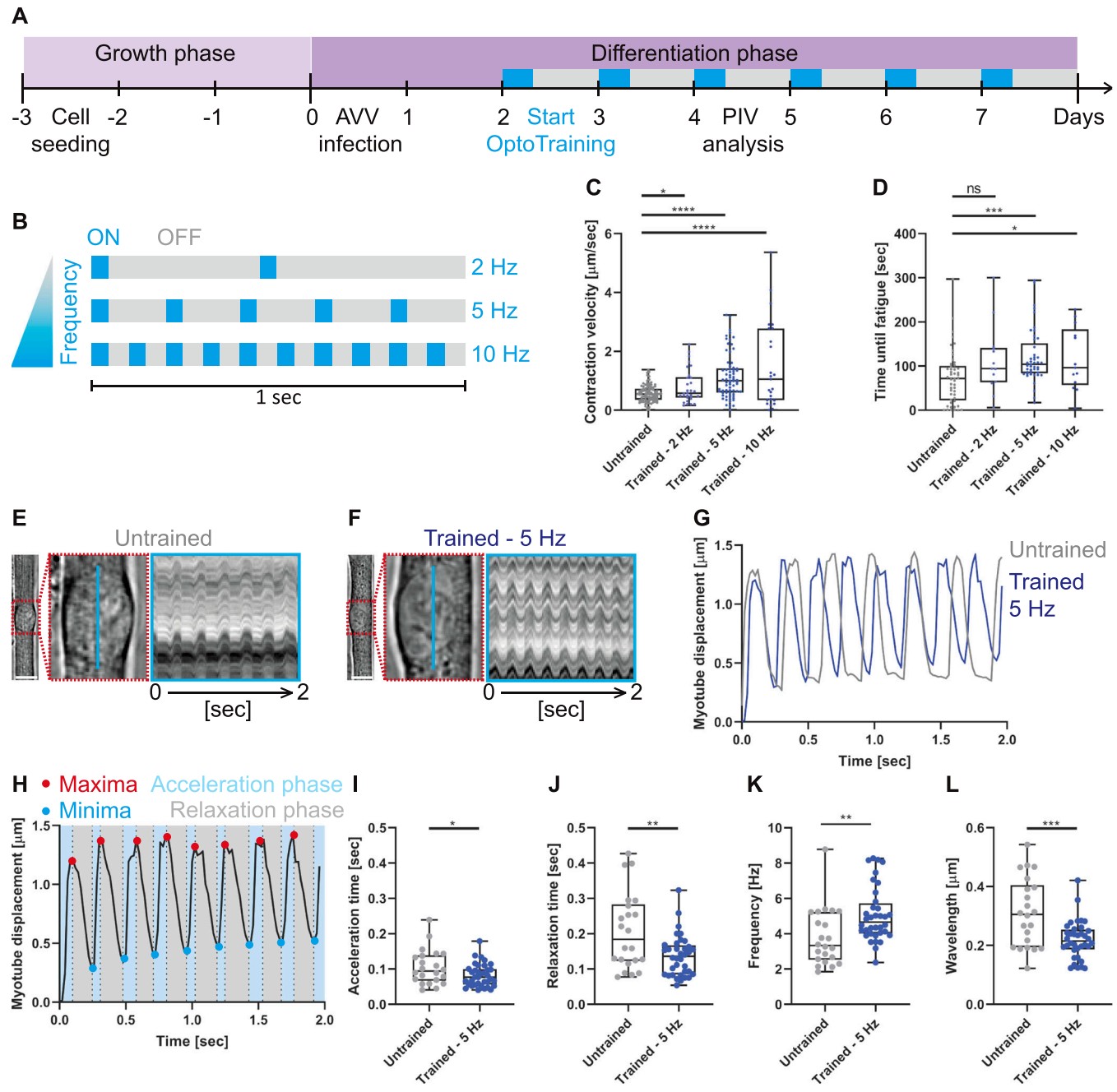

**Figure 2. Trained myotubes undergo faster contraction cycles.**
**(A)** Experimental protocol to submit in vitro primary mouse myotubes to OptoTraining. After the initial growth phase, myoblasts differentiate into multinucleated myotubes. Myotubes were infected with AAV9-pACAGW-ChR2-Venus at day 0 and submitted to OptoTraining from day 2 (8 h per day, blue boxes). **(B)** Schematic showing the three OptoTraining protocols at 2, 5 or 10 Hz. **(C)** Boxplots depicting the contraction velocities of untrained cells (n = 100) and cells trained at 2 Hz (n = 73), 5 Hz (n = 32), and 10 Hz (n = 25). **(D)** Graph showing time until fatigue of myotubes after 3 d of OptoTraining with different frequencies (untrained n = 49; 2 Hz n = 13; 5 Hz n = 45; 10 Hz n = 13). **(E, F)** Left: representative brightfield images of single myotubes with magnifications of myonuclei (red dashed boxes). **(E, F)** Right: 2-s kymographs (blue line) of untrained (E) and trained (F) myotubes when continuously stimulated with blue light. **(G)** Single-track displacement curves of untrained (grey) and trained (5 Hz light stimulation frequency: blue) myotubes. **(H)** Representative displacement curve illustrating metrics of quantitative analysis during contraction: detection of maxima (red) and minima (blue) allows to dissect the curve into acceleration (light blue) and relaxation (grey) phases. **(I, J)** Acceleration (I) and relaxation (J) time of individual myotubes averaged over contractions over 2 s. **(K, L)** Frequency (K) and wavelength (L) for untrained and trained myotubes. **(I, J, K, L)** Untrained n = 22, trained 5 Hz n = 36. Data information: all experiments were repeated at least three times. Data were plotted as box and whisker plots and outliers identified using ROUT method (Q = 1). Two-tailed unpaired t test; ns, nonsignificant; P < 0.05. Scale bars: (E, F) 10 μm.

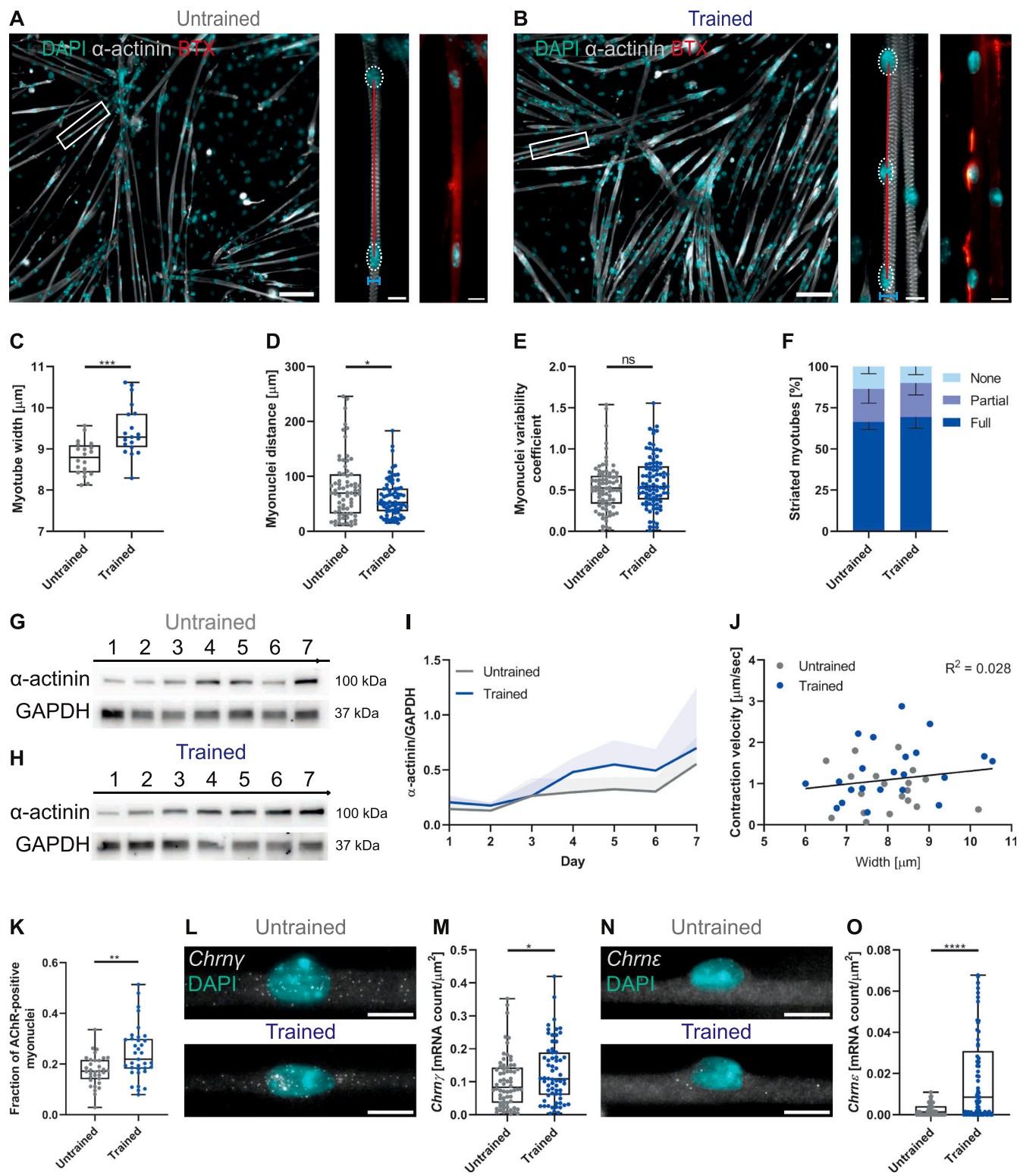

Figure 3. OptoTraining facilitates morphological maturation.

(A, B) Representative fluorescent images to assess myotube morphology of untrained (A) and trained (B) cultures (grey: α-actinin; cyan: myonuclei). Magnifications (white boxes) illustrate quantified tissue quality parameters. Left: myotube width (blue lines) and myonuclei spacing, density, and uniformity (red lines). Right: AChR clusters (red hot: BTX). (C, D, E) Boxplots showing (C) myotube width (untrained n = 20; trained n = 19), (D) myonuclei distance (untrained n = 73; trained n = 72), and (E) myonuclei variability coefficient at day 4 of differentiation (untrained n = 95; trained n = 105). (F) Percentage of striated myotubes (data summarized in grouped bar graph with mean ± SD as error bars; dark blue: fully striated; blue: partially striated; light blue: non-striated; n = 3). (G, H) Western blots of α-actinin and GAPDH (loading control) protein expression for untrained (G) and trained (H) cultures over 7 d of myotube differentiation (n = 7). (I) Graph showing temporal mean α-actinin expression

faster contractile dynamics in trained cultures are not because of thicker myotubes.

As OptoTraining promotes morphological maturation, we aimed to further validate this finding by looking at acetylcholine receptors (AChR), neurotransmitter-gated ion channels at the neuromuscular junction. During myogenesis, AChR segments increase in size and change their composition from embryonic (γ) to adult (ε) subunits (Jin et al, 2008; Cetin et al, 2020). To detect AChR in our cultures, we first stained myotubes with α-bungarotoxin (BTX) at day 4 of differentiation. We observed an increase in the number of AChR clusters around myonuclei from trained myotubes (Fig 3K), whereas we measured no difference in AChR cluster morphology (size, roundness, and circularity; Fig S4A–E). To explore further this enhanced AChR clustering, we evaluated the expression of both γ (Chrnγ-fetal) and ε (Chrnε-adult) subunits at the transcriptional level. qRT-PCR bulk analysis showed no significant differences in Chrnγ and Chrnε gene expression levels (Fig S4F). However, by using single-molecule RNA fluorescence in situ hybridization (smFISH; Pinheiro et al, 2021), we monitored RNA expression at the single-cell level of matured myotubes. We observed a higher expression of Chrnγ and Chrnε for trained myotubes (Fig 3L–O). The detection of adult AChR mRNAs further confirms enhanced maturation of trained myotubes (Bakooshli et al, 2019) but most likely does not account for the accelerated contraction dynamics considering contractions are optogenetically driven.

### Increased contractile demand triggers a down-regulation of slow-Myh and an up-regulation of fast-Myh

We hypothesized that the faster contractile dynamics after long-term training could be because of Myh switching. The pattern of Myh expression during development follows a sequential transition (Agbulut et al, 2003; Brown et al, 2012) from developmental (emb-Myh3 and neo-Myh8) to adult Myh isoforms (slow-Myh7 or fast-Myh [fast-Myh: fast-Myh2, fast-Myh1, fast-Myh4; Fig 4A]). We first investigated the expression of the developmental emb-*Myh3* and neo-*Myh8* and slow-*Myh7* genes at the single-cell level, using specific smFISH probes. Whereas the RNA levels of emb-Myh3 and neo-Myh8 do not change after long-term exercise (Fig. 4B–E), we observed a significant down-regulation of fast-Myh7 (Fig 4F and G) in trained myotubes.

As Myh7 down-regulation is characteristic of fast fiber-type specialization (Schiaffino et al, 2015), we further investigated the protein-expression patterns for distinct fiber-type–specific isoforms. First, we validated the specificity of fiber-type–specific antibodies by staining the hindlimp muscle section of 5-d-old pups and soleus cuts of 30-wk-old mice (Fig S5A and B). We then stained untrained and trained myotubes cultures for total-Myh and measured a higher fluorescent intensity for trained cultures (Fig

4H–J). To further investigate the increase in total-Myh protein expression, we examined the temporal expression pattern of total, fast-, and slow-Myh over 7 d by Western blots. Protein expression levels were normalized to α-actinin to account for sarcomerogenesis. Over the 7 d of differentiation, untrained cultures express both slow- and fast-Myh isoforms with only minor fluctuations in relative expression (Fig 4K). Consistent with the immunofluorescence, total-Myh increase starting at day 4 in trained cultures. Fast-Myh account for this increase as levels of fast isoforms are up-regulated, whereas slow myosin isoforms decrease (Fig 4L). More specifically, Western blot and qRT-PCR analysis revealed a down-regulation of slow-Myh7 and fast-Myh2 and an up-regulation of fast-Myh4 after training (Figs 4M and S6A and B), indicating a shift towards glycolytic fast fibers (Johnson et al, 2019). We next evaluated if the slow-to-fast-fiber type switch is accompanied by metabolic changes. However, we measured no changes in lactate and mitochondrial enzyme concentrations between trained and untrained cells (Fig S7A–C), suggesting that we are only recapitulating one stage of fiber-type switching with long-term in vitro exercise.

## Discussion

Fiber-type composition confers muscle the appropriate kinetics to serve their contractile demands. This is particularly relevant during development (Agbulut et al, 2003) or exercise (Qaisar et al, 2016), in which fiber types can switch. In this study, we provide an experimental approach to accelerate myotube maturation and promote fiber-type switching in a dish through in vitro exercise. Using an optogenetic setup compatible with cell culture, we induce myotubes to contract according to specified training protocols over several days. Long-term exercise of myotubes results in faster contraction dynamics and resistance to fatigue. By monitoring RNA and protein levels of developmental and adult Myh isoforms, we observed the premise of fiber-type switching in trained myotubes with the down-regulation of slow-Myh7 and the concomitant up-regulation of fast-Myh. The accelerated contraction dynamics are also accompanied by faster muscle cell maturation at both structural and transcriptional levels. Overall, we provide an in vitro strategy to model fiber-type switching during development, exercise or disease.

Submitting myotubes to mechanical stretch is a widely used approach to induce muscle contractions. Various molecular and biochemical alterations, including improved maturation, increased myotube size, and alignment, have been reported (Ren et al, 2021). Optogenetic stimulation of C2C12, which under standard culture conditions do not contract, facilitates maturation of striated, contractile myotubes (Asano et al, 2015). Using pharmacological

---

normalized to GAPDH (error bars showing SD). **(J)** Contraction velocity plotted relative to cell width of trained and untrained myotubes. $R^2$ (0.028) was computed for the whole dataset of untrained and trained myotubes via two-tailed Pearson's correlation with a 95% confidence interval (n = 41). **(K)** Box and whisker plot showing the ratio of AChR-positive myonuclei (untrained n = 33; trained n = 35). **(L, M, N, O)** Chrnγ (L) and Chrnε (N) smFISH probes (grey) and DAPI staining (cyan) in trained and untrained myotubes with plot showing (M) Chrnγ (untrained n = 68; trained n = 64) and (O) Chrnε (untrained n = 58; trained n = 68) mRNA count per $\mu m^2$ myotube area. Scale bars: (A, B) 100 $\mu$m; (A, B magnifications; L, N) 10 $\mu$m. Data information: all experiments were repeated at least three times and outliers were identified using ROUT method (Q = 1). Two-tailed unpaired *t* test; ns, nonsignificant; $P < 0.05$.

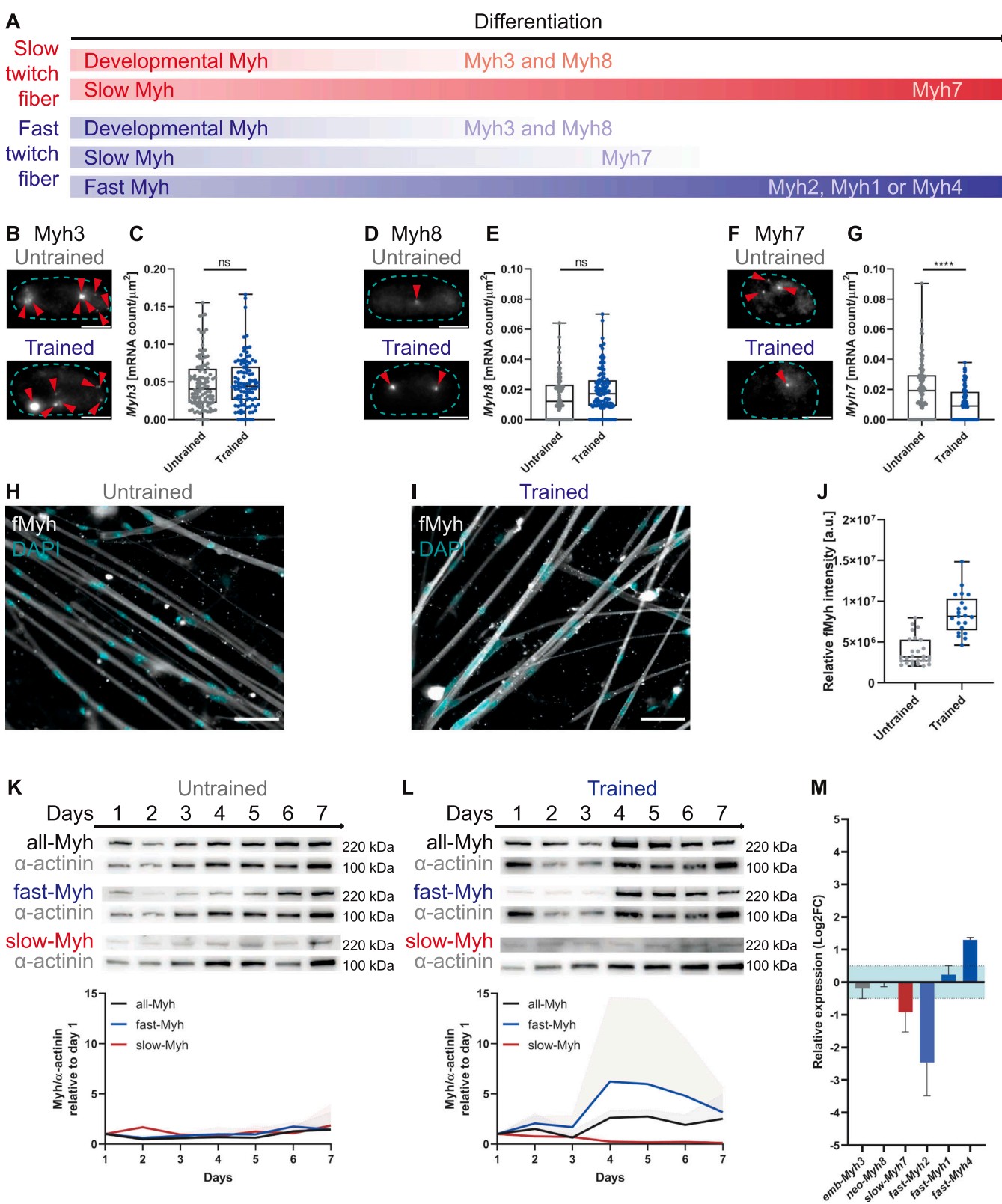

**Figure 4. Long-term mechanical training up-regulates the expression of neo-Myh8 and fMyh isoforms.**
**(A)** Simplified scheme showing temporal expression patterns of Myh isoforms for slow and fast twitch fiber types. **(B, C, D, E, F, G)** smFISH for Myh3 (B), Myh8 (D), and Myh7 (F) mRNA expressed in myonuclei (cyan dotted outline; red arrowheads highlighting smFISH probes) of untrained and trained myotubes. Boxplots of RNA expression per myonucleus area. **(C, E, G)** For (C) untrained n = 106; trained n = 98, (E) untrained n = 82; trained n = 101, and (G) untrained n = 100; trained n = 88. **(H, I)** Immunofluorescence

treatments, the development of contractility has been linked to an increase of intracellular calcium levels that promote sarcomere assembly (Sebille et al, 2017; Chapotte-Baldacci et al, 2022). To further assess developmental maturity and define fiber type, Khodabukus et al have characterized twitch dynamics of electrically stimulated 3D-engineered myobundles (Khodabukus et al, 2019). Trained muscle cells possess faster relaxation kinetics accompanied by higher force production, decreased fatigability, and increased glucose metabolism. However, no changes in glucose and mitochondria protein levels were detected, suggesting only a partial shift towards a fast fiber type.

Neuromuscular activity plays a key role in establishing fiber type during development as innervating motor neurons fire at distinct frequencies (Hennig & Lømo 1985). By mimicking specific firing patterns using electrical stimulation in vivo, a fast-to-slow or slow-to-fast fiber-type switch can be triggered using either low (Windisch et al, 1998) or high frequencies (Blaauw et al, 2013), respectively. In vitro, most studies use an intermittent optogenetic light activation at 1–20 Hz to mimic neuronal input. By stimulating myotube cultures at an intermediate frequency (5 Hz), we generated myotubes with intermediate contractile properties, namely fast twitching velocity and high resistance to fatigue. We observed a tetanus-twitching behavior for some muscle cells trained at 5 or 10 Hz, which was not observed in cells submitted to 2 Hz and in control cells. Complementary studies reported tetanus twitching at high frequencies (>8 Hz; Sebille et al, 2017; Bakooshli et al, 2019). We therefore speculate that a complete switch to fast fiber type with low fatigue resistance may require a faster stimulation frequency, found in vivo (Hennig & Lømo 1985), and longer culture times.

Improvements of in vitro muscle cultures would provide greater control over fiber-type switching. Aligning and attaching muscle cells is a strategy to prolong and mature the cultures allowing to recapitulate the full transition from fetal to adult Myhs. Other cell types could also be co-cultured to influence the expression of certain fiber type and the coordinated expression of Myh within a single myofiber. Myotubes cultured without other cell types adopt a "default path" towards a slow fiber type. However, the firing frequency of motor neurons during development determines myofiber type (Hennig & Lømo 1985), and denervation leads to uncoordinated Myh expression (Dos Santos et al, 2020). Both processes could be recapitulated in optogenetically stimulated motor neurons co-cultured with muscle. Such systems would be particularly relevant to investigate myonuclei coordination and the mechanisms of fiber-type switching at several cellular levels (genetic, structural, functional, and metabolic).

Controlling fiber type in cultured muscle systems will also be relevant for disease modelling when using patient-derived–induced pluripotent stem cells. As many myopathies preferentially affect certain fiber types (Talbot & Maves, 2016), biological investigation and drug testing should be performed accounting for fiber-type composition. It will primarily be interesting to observe if the susceptibility of certain fiber types to muscle diseases is conserved in vitro. This would provide drug development models for therapies aimed at promoting fiber switching to the less afflicted fiber type. For example, compounds found to preserve muscle integrity in Duchenne muscular dystrophy promote the less affected slow muscle phenotypes. In addition, exercise protocols can be personalized to prevent or delay fiber switching in muscle disorders worsened by fiber composition alterations.

# Materials and Methods

### Primary mouse myotubes in vitro culture

All procedures using animals were approved by the institutional ethics committee and followed the guidelines of the Portuguese National Research Council Guide for the care and use of laboratory animals. In vitro myotubes were generated as described previously from primary mice myoblasts (Pimentel et al, 2017). Myoblasts were isolated from C57BL/6J mice 5–7 d after birth. The tissue was minced and digested using 0.5 mg/ml collagenase (Sigma-Aldrich) and 3.5 mg/ml dispase (Roche) in PBS. The cell suspension was then filtered and cells were plated in IMDM (37°C, 5% $CO_2$; Thermo Fisher Scientific). After 4 h, the medium was collected containing non-adherent myoblasts. Cells were then centrifuged (240$g$, 5 min) and resuspended in growth medium (IMDM + 20% FBS + 1% chick embryo extract + 1% penicillin/streptomycin). Cells were plated onto IBIDI dishes coated with 1:100 matrigel (Corning Inc.) at a density of 200,000 cells/ml. Differentiation was initiated after 4 d by changing the medium to differentiation medium (IMDM + 10% horse serum (HS) + 1% penicillin/streptomycin). At day 1 of differentiation, a thick layer of 150-$\mu$l matrigel (1:1) was added on top of forming myotubes and the medium was supplemented with 100 ng/ml agrin. Cells were cultured up to 7 d at 37°C and 5% $CO_2$. After 4 and 7 d of differentiation, the pH was 7.5–8.

### OptoTraining

At day 0 of differentiation, cells were infected with pACAGW-ChR2-Venus-AAV9 (Addgene). 1 $\mu$l was added to 500 $\mu$l of the differentiation medium. Cells were incubated O/N at 37°C and 5% $CO_2$. Cultures were washed once with PBS, before matrigel, and differentiation medium were added as described before. At day 2, OptoTraining was initiated using the home-made OptoPlate. Trained cells were submitted to a specific temporal light pattern (50-ms light pulses at 2, 5 or 10 Hz) for 8 h per day. ChR2-expressing control cells were kept in the dark.

---

images of untrained (H) and trained (I) myotubes stained for total Myh. **(J)** Measured fluorescence intensity of total Myh per myotube (untrained n = 26; trained n = 21). **(K, L)** Western blots of total Myh (black), fast-Myh (blue), and slow-Myh (red) and α-actinin (grey) as a loading control with associated graphs for untrained (K) and trained (L) cultures. Graphs showing the relative expression profile (mean ± SD) of respective Myh isoforms normalized to α-actinin, which was used as a muscle-specific loading control (n = 3). **(M)** Graph showing mRNA expression of distinct Myh isoforms for untrained cultures relative to trained cultures, quantified using qRT-PCR (n = 3). Ct values were normalized to the expression of housekeeping gene HRPT. Light blue zone: nonsignificant fold change (cut-off value −0.5 > log2FC > 0.5). Data information: (C, E, G, J) Data plotted as box and whisker plots. Outliers were identified using ROUT method (Q = 1). Two-tailed unpaired $t$ test; ns, nonsignificant; $P < 0.05$. Scale bars: (B, D, F): 5 $\mu$m; (H, I): 50 $\mu$m.

### Fixation and immunostainings of cultured cells

Cultured cells were washed 1x with PBS and fixed for 10 min with 200 $\mu$l of 4% PFA. Subsequently, dishes were washed 2x with PBS. If needed, cells were incubated with 1:50 Alexa Fluor 488 $\alpha$-bungarotoxin (BTX; Invitrogen) in PBS for 20 min at RT. Cells were permeabilized with 0.5% tritonX100 in PBS for 5 min and blocked with 10% Goat serum in PBS with 5% BSA (blocking buffer, BB) for 1 h. Primary antibodies were diluted in 200 $\mu$l BB with 0.1% saponine and incubated overnight at 4°C. Dishes were washed with PBS 3x for 5 min under agitation. Secondary antibodies (1:400) together with DAPI (1:10,000) were incubated in BB + 0.1% saponine for 1 h and subsequently washed 3x for 5 min with PBS under agitation. 150 $\mu$l of fluoromount-G (SouthernBiotech) were added per dish and dried for 24 h at 4°C. All use antibodies are listed in Table S1. Images were acquired using an inverted widefield fluorescence microscope (Zeiss Cell Observer; 40x EC Plan-NeoFluar Air, NA 0.75, WD 0.71 mm) or a confocal microscope (Zeiss LSM 710; 63x Plan-Apochromat DIC Oil, NA 1.40, WD 0.19 mm).

### Fixation and immunostaining of muscle cryosections

Muscle from 5-d-old pups (hindlimp) or 30-wk-old adults (soleus) was wrapped in tragacanth gum and frozen in liquid nitrogen-cooled isopentane for histological assays. Transversal 6-$\mu$m cryosections were prepared with a cryostat and samples were stored at −80°C.

For immunostainings, slides were thawed to RT and fixed in 4% paraformaldehyde for 10 min. Fixed sections were washed with PBS + 0.1% tween (PBS-T) and antigen retrieval was preformed using a low pH (20 min at 95°C and PH 6). After wash with PBS-T for 30 min, nonspecific sites blocked with 5% BSA + MOM Blocking (30 $\mu$l/ml, # VB001; R&D Systems) in PBS for 2 h. The primary antibodies (Table S1) were added O/N at 4°C in 2% BSA + 5% goat serum + 0.1% triton in PBS. Cross sections were washed three times with PBS-T and then incubated with secondary antibodies together with DAPI for 2 h. After wash again in PBS-T and dry at RT, sections were mounted in Fluoromount G (#00-4958-02; Thermo Fisher Scientific). Slides were acquired using an inverted widefield fluorescence microscope (Zeiss Cell Observer; 40x EC Plan-NeoFluar Air, NA 0.75, WD 0.71 mm).

### Computational analysis

All images were processed using FIJI (Schindelin et al, 2012) and analysed using in-house MATLAB (R2019a; MathWorks) codes to extract structural parameters (average myotube width, percentage of striated myotubes, myonuclei distribution), calculating the proportion of myonuclei with high BTX intensity, and displacement curve analysis. FIJI macros for preprocessing images and MATLAB code for segmentation and calculation of metrics is available on Github (https://github.com/dhardma2/MyoChip/tree/main/Opto).

### Structural analysis of cellular maturation

Stained images were preprocessed using FIJI (Schindelin et al, 2012) to apply z-projection, subtract background, and binarize images stained with DAPI and $\alpha$-actinin. The proportional area of myotubes in each image was calculated using an in-house MATLAB (R2019a; MathWorks) code to determine the proportion of regions with $\alpha$-actinin fluorescence in binarized images. The total area of myotubes was then divided by the total length of the centerline skeletons of the myotube regions to estimate the average myotube width for each image. DAPI-stained nuclei residing within regions of $\alpha$-actinin fluorescence were designated as myonuclei. The sum of myonuclei in each image was determined using MATLAB code and normalized to calculate myonuclei per millimeter$^2$.

Myonuclei distribution was measured in MATLAB by selecting myotubes at random and manually labelling nuclei. These spatial distribution data were then used to calculate the mean distance between neighboring nuclei in each myotube. Uniformity of myo-nuclei distribution was determined using the coefficient of variation (SD in distance divided by mean distance between nuclei for each recorded myotube). The percentage of randomly selected myotubes with full, partial (<50%), and no striations were recorded manually.

### Calculating proportion of nuclei with high BTX intensity

Myotube nuclei regions are segmented from DAPI-stained images using ImageJ software. The mean value of BTX pixel intensity within each myotube nucleus region is calculated. A value for the background BTX fluorescence intensity is determined from the modal bin value of a histogram of nuclei mean BTX intensities. High-mean nuclei BTX intensity, indicating the presence of acetylcholine receptors, is defined as being any value over two times this background value. The proportion of nuclei with high-mean BTX intensity is then calculated for each image.

### Displacement curve analysis

A fast Fourier transform was applied to myonuclei displacement curves in MATLAB to determine the frequency of a representative sine wave for the motion of each cell after stimulation. To assess variation in muscle cell twitch motion over time, in-house MATLAB code was used to find the maxima and minima of single-cell displacement curves recorded over 2 s. The amplitude for each twitch was calculated and its contraction and relaxation time were computed. The code used for PIV data analysis is available on GitHub (https://github.com/dhardma2/MyoChip/Opto).

### PIV analysis

We extracted contractile parameters using PIVlab (Thielicke & Stamhuis, 2014). Because of the heterogeneity of contraction along the muscle cell, myonuclei were used as reference points to measure displacement and instantaneous velocity per myotube. The program divides images into small interrogation windows (Fast Fourier Transform window deformation, interrogation area phase 1: 64 pixel; phase 2: 32 pixel). Via maximum correlation method, the local displacement of two consecutive windows is computed. From this, the program calculates quantitative parameters (e.g., velocity, displacement) of myotube movement and displays vector and velocity magnitude fields. To assess fatigability,

myotube contractions were induced via continuous light stimulation. And the time was measured until cells stopped moving.

## Western blots

Dishes with myotubes were washed 3x with cold PBS on ice. Cells were removed from ice and 200 $\mu$l of lysate buffer (1% SDS in 100 mM TRIS–HCl, pH 8) was added to the dishes. Cells were scrapped and transferred into 1.5 ml Eppendorf. To remove matrigel, cells were spun down for 5 min at 240*g*. The supernatant was collected and stored at –80°C. For Western blots, cell lysate protein concentrations were measured using the BCA kit (Pierce). Samples were mixed with 1:4 lammeli buffer and boiled at 95°C for 5 min. The same amount of the sample (10 or 30 ng/ml) was loaded onto 4–15% pre-cast Bis–Tris gel (Invitrogen) and run at 100 V. Proteins were transferred onto nitrocellulose membrane for 75 min at 100 V. Subsequently, membranes were blocked for 1 h in blocking buffer (BB) using 5% non-fast dry milk in TBS-T (TRIS-buffered saline with 0.1% Tween 20). Primary antibodies in BB were incubated O/N at 4°C. Membranes were washed 3x with TBS-T under agitation and then incubated with HRP-conjugated secondary antibodies for 1 h at RT. Membranes were washed 3x in TBS-T under agitation, visualized using ECL reagent (Peirce), and imaged using Amersham ImageQuant 800 Western blot imaging system. Quantification was done in Image Laboratory. All used antibodies are listed in Table S1.

## Single molecule FISH

For *Chrnγ*, *Chrnε*, and *filamin C*, mRNA probes were designed to align with the coding sequence of the mRNA of interest using the Stellaris probe designer. For *Myh* isoforms, smFISH probes were designed for gene-specific mRNA sequences. All probes were coupled to Quasar570 and are listed in Table S2. The dried oligonucleotide probe was dissolved in 400 $\mu$l RNase-free water (Invitrogen) to a stock concentration of 12.5 $\mu$M. Culture dishes with myoblasts were washed with RNase-free PBS (Ambion) and fixed 10 min at RT in fixation buffer (10% formaldehyde solution, Sigma-Aldrich, in nuclease free water). Cells were washed 2 x and permeabilized O/N in 70% ethanol at 4°C. For hybridization, dishes were washed with the wash buffer (1x saline-sodium citrate, SSC; Sigma-Aldrich, in 10% deionized formamide; Ambion) for 5 min 1.25 $\mu$M smFISH RNA probe in 10% formamide, 1% dextran sulfate (Sigma-Aldrich) in 2x SSC and incubated O/N at 37°C. Myotubes were washed 2x in wash buffer (containing 50 ng/ml DAPI at second wash) for 30 min at 37°C. 1 ml 2x SSC was incubated for 5 min at RT. Cells were then mounted using 150 $\mu$l Vectashield Antifade Mounting Medium (Vector Laboratories) and stored at 4°C. Cells were imaged within 1 wk using an inverted widefield fluorescence microscope (Zeiss Cell Observer). All images were processed in Fiji. Z-stacks of myofibers were projected to maximum intensity, the background was subtracted, and images were binarized. For *Chrnγ*, *Chrnε* and *filamin C*, mRNA signals were counted within the perinuclear region (myonuclei ±50 $\mu$m along the myotube). For myh isoforms, the area per myonuclei was computed.

## Quantitative real-time PCR (qRT-PCR)

RNA was extracted from cultured cells using TRIzol reagent (15596018; Invitrogen). RNA concentration was measured using Spectrophotometer Nanodrop 2000 (Thermo Fisher Scientific) and samples were stored at –80°C. 1,000 ng of isolated RNA were reverse-transcribed using High-Capacity RNA-to-cDNA kit (4387406; Applied Biosystems) according to the manufacturer's instructions. cDNA samples (20 ng/ml) were stored at –20 degrees. Real-time PCR was performed using Power SYBR Green PCR Master Mix (4367659; Applied Biosystems). The efficiency of all designed primers (annealing temperature 58–62°; see Table S3) was validated with a standard curve assay using 32, 8, 2, and 0.5 ng of cDNA. Ct values for each sample were normalized to the housekeeping gene HPRT. In accordance with MIQE guidelines (Bustin et al, 2009), we calculated relative expression levels (2–$\Delta\Delta$Ct).

## Lactate assay

The supernatant and cell lysate were collected at day 7 of differentiation. The lactate concentrations were measured using the Lactate-Glo Assay (J5021; Promega). The assay was performed following the manufacturer's guidelines. The luminescence was measured using a Spark Multimode Microplate Reader (Tecan).

## Contractility assay and PIV analysis

To determine functional properties of untrained and trained cells, we measured myotube contraction velocity and fatigability at day 4 of maturation. To do so, we performed time-lapse imaging with a high-temporal resolution (20 ms/frame) using a Zeiss Cell Observer SD microscope (63x oil immersion objective Plan-Apochromat 63x, NA M27). Single-striated myotubes with nuclei at the periphery were chosen. First, the cells were imaged without photostimulation. Subsequently, the cells were submitted to continuous blue light illumination (470 nm). We computed contractile parameters over time periods of 2 s in MATLAB using PIVlab, an image-based PIV analysis software (Thielicke & Stamhuis, 2014), detailed in supplementary information.

## Statistical analysis

Statistical analysis was carried out in GraphPad Prism (using unpaired parametric *t* test). The distribution of data points is expressed as mean ± SD for three or more independent experiments. Outliers were identified using ROUT method (Q = 1).

# Data Availability

Data will be available upon request.

## Supplementary Information

## Acknowledgements

We thank all partners of the MyoChip consortium and all members of the Edgar Gomes and the Claudio Franco laboratory. We thank Afonso Malheiro and Judite Costa for support and discussions. We thank António Temudo, Ana Nascimento, Aida Lima, and José Rino from the Bioimaging facility at iMM for imaging assistance. This project has received funding from the European Union's Horizon 2020 research and innovation program under grant agreement FET-OPEN No. 801423 and the European Research Council H2020-GA 810207-ARPCOMPLEXITY, the ERC-2022-POC2 n 101113328 and the Association Française contre les Myopathies (AFM). W Roman was supported by the Baker Foundation of Australia.

### Author Contributions

K Hennig: conceptualization, data curation, software, formal analysis, validation, methodology, and writing—original draft, review, and editing.

D Hardman: data curation, software, and writing—review and editing.

DMB Barata: conceptualization, methodology, and writing—review and editing.

IIBB Martins: methodology and writing—review and editing.

MO Bernabeu: supervision and writing—review and editing.

ER Gomes: resources, supervision, funding acquisition, validation, and writing—review and editing.

W Roman: conceptualization, supervision, funding acquisition, investigation, visualization, and writing—original draft, review, and editing.

### Conflict of Interest Statement

The authors declare that they have no conflict of interest.

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
