## [Reviewer comments · Life Science Alliance]

Life Science Alliance

Generating fast-twitch myotubes in vitro with an optogenetic-based, quantitative contractility assay

Katharina Hennig, David Hardman, David Barata, Ines Martins, Miguel Bernabeu, Edgar Gomes, and William Roman DOI: <https://doi.org/10.26508/lsa.202302227>

Corresponding author(s): William Roman, Australian Regenerative Medicine Institute

Review Timeline:

Submission Date:	2023-06-19
Editorial Decision:	2023-06-22
Revision Received:	2023-06-22
Editorial Decision:	2023-07-14
Revision Received:	2023-07-26
Accepted:	2023-07-27

Transaction Report:

June 22, 2023

Re: Life Science Alliance manuscript #LSA-2023-02227

Mr. William Roman
Instituto de Medicina Molecular
Unknown
Portugal

Dear Dr. Roman,

Thank you for submitting your manuscript entitled "Generating fast-twitch myotubes in vitro using an optogenetic-based, quantitative contractility assay" to Life Science Alliance. We invite you to re-submit the manuscript, revised according to your Revision Plan.

Thank you for this interesting contribution to Life Science Alliance. We are looking forward to receiving your revised manuscript.

Sincerely,

B. MANUSCRIPT ORGANIZATION AND FORMATTING:

Full Revision

Manuscript number: #RC-2022-01697

Corresponding author(s): William Roman; Edgar R. Gomes

[Please use this template only if the submitted manuscript should be considered by the affiliate journal as a full revision in response to the points raised by the reviewers.]

*If you wish to submit a preliminary revision with a revision plan, please use our "Revision Plan" template. **It is important to use the appropriate template to clearly inform the editors of your intentions.**]*

1. General Statements [optional]

This section is optional. Insert here any general statements you wish to make about the goal of the study or about the reviews.

We would like to thank the reviewers for their careful evaluation of our study. The goal of this work is to demonstrate that fiber type composition can be altered with exercise of *in vitro* muscle cultures. These findings provide an additional strategy to better mimic muscle *in vitro* for biological investigation and disease modelling. The reviewers' comments strengthened the conclusions of our study.

In this point-by-point answer, we also include a statement on the status of each comment based on work we have performed since receiving the reviews.

This section is mandatory. Please insert a point-by-point reply describing the revisions that were already carried out and included in the transferred manuscript.

Reviewer #1 (Evidence, reproducibility and clarity (Required)):

The manuscript by Henning et al describes a method to induce myofiber subtype specification in vitro based on optogenetics and particle image velocimetry. The work is well performed and the manuscript is clear. The findings might be useful to the muscle community, but there are some issues which should be addressed in order to improve the quality and impact of the manuscript.

My main concern is that the whole work is performed in murine cells. Although I appreciate that the authors have used primary myoblasts rather than cell lines, I also think that the key advantage of such in vitro platforms is the possibility to "humanise" the experiments as much as possible. In this context, the key findings of this work should be reproduced using human myoblasts. This will significantly enhance the relevance of the work.

Full Revision

Point 1.1) We thank the reviewer for the suggestion. We performed several pilot experiments to “humanize” our findings. We infected hiPSC-derived myotubes (van der Wal et al., 2018) and human immortalized myotubes (Mamchaoui et al., 2011) with either AAV9-pACAGW-ChR2-Venus-AAV or AAV1-pACAGW-ChR2-Venus-AAV to maximize infection efficiency. Unfortunately, human immortalized myotubes did not express ChR2 with either virus serotype, not permitting optogenetic training on these cultures. For hiPSC-derived myotubes, the infection rate was improved but insufficient to evaluate the effect of long-term intermittent light stimulation across the cultures. Moreover, the contractile behavior of hiPSC-derived myotubes expressing ChR2 significantly differed from primary mouse myotubes. Cells underwent a single and slow contraction when compared to the cyclic contractions observed in mouse myotubes. This suggests that the maturation of the contractile apparatus of 2D hiPSC-derived myotubes is insufficient to perform consistent *in vitro* training studies.

As such, we agree with the reviewer that reproducing our key findings with human cells would improve the relevance of this work. However, due to the experimental limitations described above, significant improvements in human myotube maturation *in vitro* are required to perform such experiments. Our work is a proof of principal that fiber type composition can be influenced *in vitro* through contraction stimulation. We expect these findings to be translated to human cultures when the field has discovered the necessary protocols to push human myotube maturation.

Status: **Failed due to technical limitations**

Other issues:

1) *From a methodological perspective, I think some clarifications are needed on the western blots shown in Fig 4K-L, as the pattern of Myh3 and Myh8 in both panels appear very similar. This could easily be ruled out by providing raw data/images. Please accept my apologies if this is simply caused by similar migration patterns in the gels (worth checking).*

Point 1.2) The very similar appearance of both patterns is due to the same molecular weight (220 kDA) of distinct myh isoforms. After an initial staining of western blot membranes, primary and secondary antibodies were stripped off and the membrane was subsequently re-probed using a primary and secondary antibody. We incubated stripped membranes with secondary antibodies only and observed no signal, confirming the stripping was efficient. We have updated the representative images of the Western Blot membranes in Fig. 4K,L and Fig. S6 and included the α -actinin loading controls on which the bands are normalized to account for sarcomerogenesis. As stated in the manuscript, raw data will be available upon demand.

Status: **Completed**

2) *Figure 3K-L (BTX): better imaging should be performed to assess morphology of NMJ (eg. pretzel-shaped as in mature/adult NMJ?)*

Point 1.3) We agree with the point raised by the reviewer. We performed a quantitative morphological evaluation of BTX stainings (Acetylcholine receptors) imaged with a confocal

Full Revision

microscope (Fig. 4SA-E). Additionally, we computed the number of BTX positive myonuclei. We measured no significant difference in NMJ morphology (size, roundness and circularity) but observed a 1.5-fold increase in the number of AChR clusters per myonucleus in trained myotubes (Fig. 3K). In general, a morphological assessment of the NMJ is difficult in this *in vitro* system due to our inability to generate mature muscle end plates with pretzel shapes as seen in *in vivo* adult NMJs.

We have included the new data into our manuscript (Fig. 3K; Fig. S4A-E).

Status: **Completed**

3) *Figure 3 N-P: Why did the authors used a relatively complex techniques such as smFISH to answer a question more simply addressable with more conventional (and perhaps less operator dependent) techniques such quantitative PCR?*

Point 1.4) qPCR analysis of *ChrnE* and *ChrnG* displayed insignificant expression changes between trained and untrained cultures despite exhibiting some trends. We reasoned that bulk approaches such as qPCR could abate differences due to the heterogeneity of our primary myotube cultures (presence of non-muscle cell types and varying degrees of muscle cell maturation). We therefore opted to monitor AChR expression at the single cell level in mature muscle cells, similar to those selected to perform the contraction analysis.

To better reflect this process, we inserted the qPCR data of *ChrnE* and *ChrnG* followed by the smFISH data in the manuscript. The new data was included in the supplementary information (Fig. S4 F).

Status: **Completed**

Reviewer #1 (Significance (Required)):

Nature and significance: as mentioned in the previous section, the work can be very significant if expanded to human myoblasts/myotubes, which can have different slow/fast myosin expression pattern. The work is clearly methodological/descriptive, so showing an application of this technique using diseased/mutant cells may increase its relevance even more (but I do not believe it is a key barrier to publication).

We thank the reviewer for his comments as the “other issues” raised will significantly improve the manuscript. With regards to using human myotubes, we were unable to apply long-term *in vitro* exercise in human cultures due to technical limitations. Nevertheless, it is our opinion that the main contribution of this manuscript is to show that fiber switching can be achieved *in vitro* and that this could be routinely used in the next generation of human *in vitro* muscle systems.

Comparison with other methods: Similar methods have been published but not with this level of resolution.

Full Revision

Expertise: muscle disease and regeneration, in vitro and in vivo models.

Reviewer #2 (Evidence, reproducibility and clarity (Required)):

The work presented shows that muscle stem cells isolated from 5-day-old mice can be transduced with a DNA coding for a Channelrhodopsin2-Venus which will allow the muscle cell to be excited by a light beam (475nm) and to induce the contraction of myotubes. The authors measure the speed of contraction, relaxation and fatigability of such cells as a function of a more or less long excitation time. In particular, they show that myotubes in culture, excited at a frequency of 5 Hz, 8 hours per day for 7 days are larger than unstimulated myotubes and are more resistant to fatigue. Surprisingly, they show that myotubes stimulated at the low frequency of 5Hz express the neonatal Myosin heavy chain more than the slow Myh whose expression is known in adult muscle to be specifically strong in muscle fibers stimulated at low frequency. As the authors do not apply a high stimulation frequency (100Hz) to their culture, it is difficult to conclude whether the stimulation frequency applied in the study induces a specific phenotypic specialization of the myofiber, or a more general role. In this respect, the size of the myotubes obtained after training seems to be increased, showing a hypertrophic effect on the cultured myotubes. This study does not allow us to conclude, beyond the expression of the Myh8 gene, on the “gain” of the fast-twitch specialization of the myofiber by repeated stimulation over several days. A complementary study would certainly provide elements to better understand the role of muscle fiber stimulation, apart from the trophic contribution provided in vivo by the motoneuron. If the study is well conducted, some points are nevertheless important to address before publication.

Reviewer #2 (Significance (Required)):

- Figures 4F/G are difficult to understand: the Myh7 signal seems much higher in trained myonuclei (F), but the histogram shows the opposite (G).

Point 2.1) We apologize for the confusion. The apparent higher Myh7 signal in trained cells in Fig. 4F is due to background noise in the image. When mRNA is expressed, the smFISH probes are visible as fluorescent puncta. For clarity, we updated the representative images for the smFISH probes and highlighted the smFISH puncta with arrows. We also adapted the y-axis of each graph to better represent the analysis of mRNA counts.

Status: **Completed**

- Figures 4L, the western blot shows the same increase in Myh3 and Myh8 at day 4, while the graph shows an increase at d4 only in Myh8, why?

Full Revision

Point 2.2) We now provide a different western blot to better reflect the quantification. It is important to note that we have normalized the band intensity to α -actinin instead of a house keeping gene to account for changes in sarcomerogenesis over the lifetime of the cultures. As such, although we observe an increase in Myh3 intensity, it is counter balanced by an increase in α -actinin expression. We have now added the α -actinin bands to all western blot images (Fig. 4K,L and Fig. S6).

Status: **Completed**

- For immunocytochemistry against fMyh (Fig4 H, I) as well as for Western blots (Fig 4M, N), the authors have to provide arguments regarding the specificity of the antibodies used: some fMyh-specific antibodies recognize, Myh 3, 8, 1, 2, and 4, some only Myh 8, 1, 2, and 4, so it is quite difficult to conclude on the experiments using sc-32732 antibodies, (clone F59) which Myh are actually recognized in Western blot or immunocytochemistry.

Point 2.3)

We evaluated used myh antibodies by staining neonatal (5 days old pups) and adult muscle sections (TA, EDL, and Soleus). We indeed observed some issues with antibody specificity as rightfully highlighted by the reviewer. The antibody sc-32732 (Santa cruz) stained all fiber types contrary to the manufacturer's notice. We therefore refer to these stainings as "total-Myh" in the manuscript. Antibody PA5-72846 (Thermo Fisher Scientific) is specific to neonatal (Myh8) but also all fast myh isoforms (Myh2, 1, and 4), termed "fast-Myh" in the article. For consistency, the slow-Myh7 antibody (A4.951; DSHB), which is specific for slow fibers, was renamed "slow-Myh". Despite these changes, our conclusions remain similar as we observe an increase in fast-Myh isoforms. To identify which fast isoforms account for the upregulation in fast_Myh, we performed western blots for Myh1, Myh2 and Myh4 (Fig. S5) as well as qPCRs (Fig. 4M) in trained and untrained cultures. We mostly observe a decrease in slow-Myh and an increase in fast-Myh4 at both protein and mRNA levels, confirming that long term in vitro exercise promotes a slow to fast myosin transition.

In the manuscript we have summarized our main findings for the antibodies that were used (sc-32732, PA5-72846, A4.951) in figure (Fig. S5), adapted the names, and updated Fig. 4K,L.

The conclusion of a fiber type switch due to *in vitro* exercise was confirmed with qPCR (Point 2.4) and do remain valid. We greatly thank the reviewer for his expertise in the use of these antibodies.

Status: **Completed**

*While 10Hz stimulation is known in vivo to increase the slow program, and Myh7 expression in adult muscles, the authors show that ex vivo this is not the case with primary myotubes, with Myh7 protein level not being upregulated in the 7 day stimulation paradigm, while on the contrary Myh8 expression is upregulated. I think it would be important to quantify the **mRNA of each of the Myh genes** to be sure that there is no problem with the antibodies, which could recognize several Myh proteins, in the absence of a resolving acrylamide gel allowing visualization and relative level of each isoform according to its migration. Nevertheless, this is an interesting*

Full Revision

observation that could be related to the early phases of muscle contraction in vivo. Indeed, it has been shown in rats that early postnatal development animals are essentially sedentary and whose muscles (Sol and EDL) are stimulated by short intermittent bursts similar to 10Hz (doi: 10.1111/j.0953-816X.2004.03418.x) during the first 2-3 weeks of life. This should be compatible with Myh8 expression. It would be relevant in this idea to verify that the paradigm presented leads to myotubes with a "neonatal" phenotype. Quantification of the expression level of genes specifically expressed during the neonatal period, compared with those expressed in adult slow or fast myofibers, would enhance the conclusions drawn by the authors.

Point 2.4) The reviewer raises an important technical limitation of observing Myh proteins to identify fiber types due to the cross-reactivity of antibodies. Despite our best efforts to select and validate the appropriate antibodies, we agree that investigating mRNA expression of individual Myh isoforms would strengthen the conclusion of our study. We therefore designed specific primers and performed qPCR for distinct Myh isoforms on untrained and trained cultures. Using qPCR, we confirmed a fast-fiber type switch, as we observed a downregulation in slow-Myh7 and fast-Myh2 and an upregulation in fast-Myh1 (non-significant) and fast-Myh4 gene expression. Showing a trend in sequential fiber-type switching [slow-Myh7 → fast-Myh2 → fast-Myh1/4]. We have integrated this data in Fig. 4M.

With regards to the "neonatal" phenotype of these *in vitro* cultures, this does indeed seem to be the case as the cultures mostly express developmental Myh3 and Myh8 isoforms but start to express the adult myosins (see Point 2.6 for more detail).

Status: **Completed**

Should we also be cautious about bulk analysis since, as shown in Figure S1, not all myotubes express ChR2?

Point 2.5) Although 10% of myotubes do not express ChR2, we believe that 90% of infected myotubes is sufficient for bulk analysis. We nevertheless combine in our study bulk analysis with single cell assays such as smFISH and immunofluorescence, which are in line with the bulk analyses.

Status: **Completed**

May the authors correlate the ex vivo neonatal phenotype observed with the neonatal muscles they used to prepare myogenic stem cells?

Full Revision

Point 2.6) We understand from this that the reviewer would like us to check the expression of distinct Myh isoforms in our *in vitro* system and compare it to neonatal muscle. For our primary cultures, we isolate myoblasts from the soleus, gastrocnemius, plantaris and quadriceps. Our qPCR data shows that Myh3 and Myh8 are the most highly expressed isoforms but adult myosins isoforms are present (see graph below). According to Agbulut et al. JBC 2003, which compared differential expression of Myh isoforms at different times after birth and in different muscles, our cultures compare to day 7 pups, which is similar to the age from which we isolate primary myoblasts.

qPCR performed on untrained, 4-day primary cultures for Myh genes and normalized on HPRT housekeeping gene. Note that lower Ct values reflect higher gene expression.

Status: **Completed**

We thank the reviewer for his suggestions, which we have addressed. Those ensuring antibody specificity were particularly relevant to improve the manuscript.

Reviewer #3 (Evidence, reproducibility and clarity (Required)):

Summary:

In this work, the authors propose an in vitro model describing a strategy to alter fiber type composition of myotubes with a long-term, intermittent mechanical training. The authors present a model of myotubes transfected with an adenovirus, which makes them photosensitive; in this way, fibers contraction can be induced upon stimulation with blue LEDs.

Even though ChR2 expressing myotubes have previously been used by other groups (Asano T, Ishizua T, Yawo H. Optically controlled contraction of photosensitive skeletal muscle cells. Biotechnol Bioeng. 2012 Jan;109(1):199-204), no one has ever used it in the way proposed by

Full Revision

the authors. For this reason, this work opens new perspectives on the possible use for clinical and therapeutic purposes for this in vitro muscle system.

Major comments:

I believe that the authors have presented their results, conclusion and methods in a fair and clear way, so that the experiment could also be reproduced.

However, I think there are some adjustments that could be done in order to improve and strengthen the quality of this work:

- The authors have analysed the expression of different myosin heavy chain isoforms, both regarding the slow and fast twitch fibers. Though, I think it would be interesting to investigate also the expression of Myh4, which is mainly expressed in type IIB fast twitch fibers;

Point 3.1) We agree with the reviewer's comment. We added the analysis for Myh 4 (western blots and qPCR) to our manuscript (Fig. 4M and Fig. S6). We have observed an upregulation of fast-Myh4 for trained myotubes. All new findings were discussed within the main text of the manuscript.

Status: **Completed**

The authors have observed a switch in the fiber type upon prolonged intermittent stimulation with blue LEDs, which translates into a higher number of type II fibers. It is known that exercise helps rescuing the loss of type II fibers, which is typical of age-related physiological processes, such as sarcopenia (Brunner F, Schmid A, Sheikhzadeh A, Nordin M, Yoon J, Frankel V. Effects of aging on Type II muscle fibers: a systematic review of the literature. J Aging Phys Act. 2007 Jul;15(3):336-48). However, I believe that providing a deeper analysis of the metabolism of the type II fibers (i.e. oxidative or glycolytic) could be helpful in order to have a clearer view on the specific subset of fibers that are generated with the given experimental conditions;

Point 3.2) We performed lactate measurements in cell lysate and supernatant to monitor a switch from oxidative to glycolytic metabolism. We observed a non-significant increase in the lactate concentration from day 4 ($2,92 \pm 0,84$ M) to day 7 ($3,89 \pm 0,33$ M) of differentiation in the supernatant. However, we did not measure a significant difference between untrained and trained cultures. Additionally, we assessed protein level expression of oxidative phosphorylation complexes in mitochondria using western blots. We did not observe an altered level of assembly due to exercise. We have added the lactate measurements and OxPhos protein expression data in the supplementary information of the manuscript (Fig. S7).

We used specific inhibitors of the glycolytic pathway (2-DG, UK5099, Rotenone and AntimycinA) as a control to prevent trained cells from shifting towards a fast fiber type. All inhibitors were tested at different concentrations and over distinct time periods (hours to days). In general, treated myotubes either became apoptotic or lost their ability to contract, therefore we were not able to perform our contractility-based mechanical training. The preliminary data was not included in the manuscript.

Full Revision

We hypothesize that the training-induced increase in contraction speed is primarily due to the expression of fast-myh isoforms. No metabolic changes were observed after 7 days of differentiation. However, we observed a non-significant trend of decreased protein expression for mitochondrial enzymes. To trigger a full fiber-type switch towards glycolytic myotubes, cells may need to be stimulated for longer culture periods.

Status: **Completed**

Minor comments:

The text and the figures are clear and well written, and help to explain better the experimental setup and procedures. Still, I would suggest some minor adjustments:

- I would suggest providing more information on the pH used for the experiments, since it plays a pivotal role in regulating myosin ATPase activity and, thus, muscular contractility. This would improve the replicability of your experiment.

Point 3.3.) We thank the reviewer for this comment. We measured the pH at day 4 and 7 in medium of trained and untrained cultures. In both cases, pH was between 7.5-8. We added the information regarding the pH to the method and materials section “Primary mouse myotubes *in vitro* culture” in the supplementary information.

Status: **Completed**

The caption of Figure 1 is missing a description of panel E, even if it has been addressed in the text.

Point 3.4.) We apologize for this mistake. We added the missing description of Fig. 1E.

Status: **Completed**

Reviewer #3 (Significance (Required)):

This model opens new perspectives on in vitro muscle systems for the study of pathologies. The authors have been able to assess that myofibers contraction is able to induce a shift towards type II fibers, reproducing in vitro what is also known in vivo. For this reason, I believe that this model could be useful for further clinical approaches. It is important, though, to keep in mind that muscular disorders are not all characterized by a loss of type II fibers; for instance, myotonic dystrophies type I and type 2 exhibit similar phenotypes, even if different types of muscle fibers are affected.

For this reason, it would be interesting to investigate the versatility of this model in terms of giving rise to different fiber types.

Full Revision

Point 3.5.) We added a sentence in the introduction that highlights an example of muscle disorders in which slow muscle fibers are predominately affected. Concerning the versatility of the model, we added a paragraph to the discussion elaborating on how different stimulus frequency and durations could influence the specialization of fiber types.

Status: **Completed**

Overall, the reviewer's comments greatly improved the manuscript. We were disappointed not to observe metabolic changes accompanying Myh switching but hope to revisit this concept in cultures with longer life spans.

Summary of changes made in the manuscript.

Point 1.2: *pattern of Myh3 and Myh8 in both panels appear very similar* - We updated the representative images and included the loading controls in Fig. 4K,L and to Fig. S6.

Point 1.3: *Morphological assessment of NMJ* – We have computed the number of AchR cluster around myonuclei (Fig. 3K) and performed a quantitative morphological analysis of NMJs (Fig. S4 A-E).

Point 1.4): *qPCR for ChrnG and ChrnE* – We performed qPCR to evaluate the gene expression of ChrnG and ChrnE and included the data in Fig. S4 F.

Point 2.1: *Figures 4F/G: representative images of Myh7 smFISH probe and the graph showing opposite trends* – We updated the representative images of Fig. 4F and we have changed the x-axis of the graph in Fig. 4E and G.

Point 2.2: *western blot of Myh3 and Myh8 at day 4* – we have updated the representative western blot images and added the loading controls (Fig. 4K,L and Fig. S6).

Point 2.3: *validate specificity of fMyh antibody* – specificity of Myh antibodies was tested by staining neonatal and adult muscle sections. After some issues with specificity were discovered, we adapted the names and our conclusions throughout the whole manuscript. We summarized our main findings in Fig. S5 and updated Fig. 4K,L.

Point 2.4: *Validate alterations of Myh isoform expression with qPCR* – we performed qPCR experiments for all Myh isoforms (Myh1, 2, 3, 4, 7, and 8) and added our results as Fig. 4M.

Point 3.1: *investigate the expression of Myh4* – we have performed western blots and qPCR experiments to investigate Myh4 protein and gene expression. New findings were integrated in the main text of the manuscript and in Fig. 4M and Fig. S6.

Point 3.2: *providing a deeper analysis of the metabolism of the type II fibers (i.e. oxidative or glycolytic)* – we have performed lactate measurements and investigated the expression of mitochondrial enzymes via western blot, the new data can be found in Fig. S7 and was discussed in the main manuscript.

Point 3.3: *provide information of the pH* – we added the pH of the media of untrained and trained cells to the materials and methods section “Primary mouse myotubes in vitro culture” in the supplementary information.

Full Revision

Point 3.4: *caption of Figure 1 is missing a description of panel E* – We have added the missing description to the manuscript (Fig. 1E).

Point 3.5: *muscular disorders are not all characterized by a loss of type II fibers* – we have added an example of a muscle disorder, in which slow fibers are predominantly affected, to the introduction of the manuscript.

investigate the versatility of this model in terms of giving rise to different fiber types – we added a paragraph to the discussion elaborating on how different stimulus frequency can lead to different fiber types.

We have updated the Materials and Methods section to include all newly performed assays. The manuscript, figures, legends and supplementary information were updated following the formatting guideline of Development. All new findings have been integrated and discussed in the text of the main manuscript and representative figures have been added.

2. Description of analyses that authors prefer not to carry out

Please include a point-by-point response explaining why some of the requested data or additional analyses might not be necessary or cannot be provided within the scope of a revision. This can be due to time or resource limitations or in case of disagreement about the necessity of such additional data given the scope of the study. Please leave empty if not applicable.

Point 1.1: *Reproducing our key findings with human cells* – we ran pilot experiments on immortalized human cell lines and human iPSC-derived myotubes but were not able to mature these cells sufficiently nor infect them to allow long-term *in vitro* training. Increased maturation of myotubes derived from hiPSCs is an endeavor currently undertaken by many laboratories. Although we agree that reproducing our key findings in human cells would increase the relevance of this manuscript, we believe the technical limitations are too important to address this point.

July 14, 2023

RE: Life Science Alliance Manuscript #LSA-2023-02227R

Mr. William Roman
Instituto de Medicina Molecular
Portugal

Dear Dr. Roman,

Thank you for submitting your revised manuscript entitled "Generating fast-twitch myotubes in vitro with an optogenetic-based, quantitative contractility assay". We would be happy to publish your paper in Life Science Alliance pending final revisions necessary to meet our formatting guidelines.

- please upload your main manuscript text as an editable doc file
- please upload all figure files as individual ones, including the supplementary figure files; all figure legends should only appear in the main manuscript file
- please add ORCID ID for the corresponding author--you should have received instructions on how to do so
- please add the Twitter handle of your host institute/organization as well as your own or/and one of the authors in our system
- please note that the titles in the system and on the manuscript file must match
- please consult our manuscript preparation guidelines <https://www.life-science-alliance.org/manuscript-prep> and make sure your manuscript sections are in the correct order
- please add your central, supplementary figure, and table legends to the main manuscript text after the references section;
- please use the [10 author names et al.] format in your references (i.e., limit the author names to the first 10)
- please note that author names in the system and on the manuscript file should match (Inês Isabel Baltazar Belo Martins vs. Ines Isabel Martins)
- please add callouts for Figures 4I, J, N and S5A-B to your main manuscript text
- please add sizes next to all blots
- please incorporate the materials and methods section from the Supplemental Material file into the main Materials and Methods section

A. FINAL FILES:

B. MANUSCRIPT ORGANIZATION AND FORMATTING:

Sincerely,

Reviewer #2 (Comments to the Authors (Required)):

The authors have greatly improved the manuscript and responded to all comments and suggestion. I don't have further points to be addressed and agree with publication.

July 27, 2023

RE: Life Science Alliance Manuscript #LSA-2023-02227RR

William Roman
Australian Regenerative Medicine Institute
15 Innovation walk
Clayton, Victoria 3800
Australia

Dear Dr. Roman,

Thank you for submitting your Research Article entitled "Generating fast-twitch myotubes in vitro with an optogenetic-based, quantitative contractility assay". It is a pleasure to let you know that your manuscript is now accepted for publication in Life Science Alliance. Congratulations on this interesting work.

DISTRIBUTION OF MATERIALS:

Again, congratulations on a very nice paper. I hope you found the review process to be constructive and are pleased with how the manuscript was handled editorially. We look forward to future exciting submissions from your lab.

Sincerely,
